# Persistent Post-COVID-19 Syndrome in Hemodialyzed Patients—A Longitudinal Cohort Study from the North of Poland

**DOI:** 10.3390/jcm10194451

**Published:** 2021-09-28

**Authors:** Aleksander Och, Piotr Tylicki, Karolina Polewska, Ewelina Puchalska-Reglińska, Aleksandra Parczewska, Krzysztof Szabat, Bogdan Biedunkiewicz, Alicja Dębska-Ślizień, Leszek Tylicki

**Affiliations:** 1Department of Nephrology Transplantology and Internal Medicine, Medical University of Gdańsk, 80-210 Gdańsk, Poland; aleksanderoch@gumed.edu.pl (A.O.); ptylicki@gumed.edu.pl (P.T.); kpolewska@gumed.edu.pl (K.P.); bogdan.biedunkiewicz@gumed.edu.pl (B.B.); adeb@gumed.edu.pl (A.D.-Ś.); 27th Naval Hospital in Gdańsk, 80-305 Gdańsk, Poland; e.puchalska@7szmw.pl (E.P.-R.); puchola@gmail.com (A.P.); k.szabat@7szmw.pl (K.S.)

**Keywords:** COVID-19, hemodialysis, post COVID-19, long COVID-19

## Abstract

Background: After recovery from COVID-19, patients frequently face so-called “Post-COVID-19 Syndrome” defined by clusters of persistent symptoms lasting for >12 weeks which may arise from any system in the body. The long-term health consequences of COVID-19 in maintenance hemodialyzed (HD) patients remain to be investigated. Methods: In this longitudinal cohort study we described the health consequences in HD patients requiring hospitalization due to COVID-19. They were interviewed three and six months (M3 and M6) after discharge with a series of standardized questionnaires. Results: Of 144 HD patients discharged from the 7th Naval Hospital in Gdansk, 79 participants were enrolled, 39 m (49.4%) and 40 f (50.6%) with a median age of 70.0 (64.0–76.5) and an HD vintage of 40 months (17.5–88). After discharge, 93.7% and 81% reported at least one persistent symptom at M3 and M6, respectively. The most common symptoms were fatigue or muscle weakness (60.76% and 47.04%) and palpitations (40.51% and 30.14%). Dyspnea with an mMRC scale grade of at least 1 was reported by 21.5% before infection, and by 43.03% and 34.25% at M3 and M6, respectively. A decrease in the quality of life was reported in all domains of the EQ-5D-5L questionnaire but mainly in the pain/discomfort and anxiety dimensions. Mean EQ-VAS scores were 69.05, 61.58 and 64.38, respectively. Conclusion: Our study showed that HD patients may still experience persistent symptoms six months after recovery from COVID-19, which can further reduce their already poor health-related quality of life. This study highlights the need for long-term follow-up on these patients for diagnostic and rehabilitation programs.

## 1. Introduction

Recovery from COVID-19 can usually take several weeks in patients with mild to moderate symptoms. Although some patients fully recover, several observational studies have shown that a large proportion of subjects suffer from debilitating symptoms weeks and even months after COVID-19 diagnosis; among these symptoms the most common are fatigue, shortness of breath, chest tightness, headache and muscle pain [1,2]. This so-called “long COVID-19 syndrome” is defined as persistent symptoms and/or delayed or long-term complications of SARS-CoV-2 infection beyond four weeks from the onset of symptoms, and includes subacute or ongoing symptomatic COVID-19 (4–12 weeks beyond acute COVID-19); and post-COVID-19 syndrome, which includes symptoms and abnormalities persisting or present beyond 12 weeks from the onset of acute COVID-19, and not attributable to alternative diagnoses [3]. The pathophysiology of these sequelae has been supposed to involve cellular damage, a robust innate immune response with inflammatory cytokine production, and a pro-coagulant state induced by SARS-CoV-2 infection [4].

A group with relatively high incidence of COVID-19 and associated high in-hospital mortality are maintenance hemodialysis (HD) end-stage kidney disease (ESKD) patients. According to the latest European Renal Association COVID-19 Database (ERACODA) report, their 28-day probability of death is 25% for all patients, and 33.5% for patients who were admitted into hospitals [5]. In our recent study we showed the extremely high in-hospital mortality of COVID-19 HD patients, with a fatality rate up to 43.81% in subjects over 74 years old [6]. Dialysis patients have a high rate of comorbidities, a high frailty index, and are often elderly and have lowered immunity, which puts them at risk of SARS-CoV-2 infection and a severe course of the disease. Post-COVID-19 syndrome and its impact on patients’ daily functioning, and the quality of life in HD patients, requires attention. Therefore, we aimed to evaluate the presence and dynamic of post-COVID syndrome in HD patients three and six months after the disease. We hypothesized that patients still suffer from persisting symptoms of this disease which not only limit their quality of life but also require further diagnosis and therapy.

## 2. Materials and Methods

This longitudinal cohort study was performed at the 7th Naval Hospital Dialysis Unit in Gdansk. By a decision of the health authorities, HD patients in the Pomeranian voivodeship who had been diagnosed with SARS-CoV-2 infection during the first and second wave of the pandemic were hospitalized and hemodialyzed in this unit. All adult HD patients hospitalized in the hospital with a confirmed diagnosis of COVID-19 from the 6th of October 2020 to the 28th of February 2021 were eligible. Cases were considered confirmed if they had laboratory isolation of the SARS-CoV-2 by an RT-PCR test from nasopharyngeal/oropharyngeal swabs. We excluded the following patients: (a) those who died before the follow-up interview, (b) those for whom follow-up would be difficult owing to a psychotic disorder or dementia, and (c) those we were unable to contact. Three (M3) and six months (M6) after discharge from the hospital, all patients who consented to participate in the study were telephone-interviewed by trained medical students with questionnaires investigating specific persistent or emerging symptoms potentially associated with COVID-19 and the quality of their lives. The students had extensive data extraction experience gained from previous research, and were supervised by senior academics. Ethics approval for the study was obtained at the Medical University of Gdansk (NKBBN/2014/2021). The study is part of the ‘COVID-19 in Nephrology’ (COViNEPH) project focusing on the nephrological aspects of COVID-19, in particular epidemiology, prevention, disease course and treatment.

During the telephone interview patients were asked to complete a series of questionnaires, including a self-reported symptoms questionnaire (SRSQ) according to Huang et al. [7], the modified British Medical Research Council (mMRC) dyspnoea scale; the EuroQol consisted of two components, a five-dimension five-level (EQ-5D-5L) questionnaire, and the EuroQol Visual Analogue Scale (EQ-VAS). For the SRSQ, participants were asked to report newly occurring and persistent symptoms, or any symptoms worse than before COVID-19 development at the time of the interview (Appendix A). The mMRC scale is a 5-point scale to characterize the level of dyspnea with physical activity, with scores ranging from 0–4 where 0 = I only get breathless with strenuous exercise; 1 = I get short of breath when hurrying on the level or up a slight hill; 2 = I walk slower than people of the same age on the level because of breathlessness or I have to stop for breath when walking at my own pace on the level; 3 = I stop for breath after walking 100 m or after a few minutes on the level; 4 = I am too breathless to leave the house or I am breathless when dressing [8]. The EuroQol is a validated questionnaire which has two components. The first, EQ-5D-5L, is a health state classification system with five dimensions: mobility, self-care, usual activities, pain or discomfort and anxiety or depression, where each can be described by five severity levels ranging from 1—“no problems” to 5—“unable to/extreme problems” [9]. The second EQ-VAS, is the subjective rate of overall health ranging from 0 to 100 labelled as “the worst health you can imagine” and “the best health you can imagine”, respectively [10]. In mMRC and EQ-5D-5L respondents were asked to describe the severity of problems before COVID-19 (retrospectively) and longitudinally at the time of completing the questionnaires M3 and M6 after discharge: “symptoms at this moment”. Other clinical variables: baseline demographic characteristics (sex, age), body mass index (BMI), comorbidities, length of HD treatment and data from the COVID-19 hospital admissions were obtained retrospectively from available patient records. All data was reviewed by an attending physician and was finally verified by the major investigators.

Data are presented as mean ± standard deviation or median (interquartile ranges, IQR) for continuous variables, and absolute numbers (percentages) for categorical variables. We report descriptive results, and the sample size was not based on statistical hypothesis testing. The main outcome measures were: (1) the percentage of patients with persistent of COVID-19 symptoms in SRSQ; (2) mMRC score ≥ 1 in mMRC scale; (3) the percentage of responders reporting no (not any) problem across each of the five EQ-5D-5L dimensions; (4) quality of life in the analog EQ-VAS scale. Chi-square test was used for categorical variables. T-test or Mann–Whitney U tests were used to compare continuous variables where appropriate. Differences in variables measured more than twice were assessed using analysis of variance (ANOVA) or Friedman ANOVA. *p* < 0.05 (two-tailed) was considered statistically significant. Data were evaluated using the STATISTICA (version 12.0 Stat Soft Inc., Tulsa, OK, USA) software package.

## 3. Results

### 3.1. Patients

A total of 206 adult HD patients were hospitalized with confirmed COVID-19 between 6 October 2020 and 28 February 2021. Of these, 62 (30.1%) patients died and 144 were cured and discharged. Finally, 79 participants were enrolled for the questionnaire interview at M3 after discharge. 65 patients were excluded because they did not attend the follow-up interview for several reasons, which are outlined in Figure 1. Fifteen early deaths were recorded within three months after discharge and four more in the following three months. The excluded patients did not differ from the study group in terms of age: 68.5 (57–80.75) years and gender (51.43% males). The median age of the enrolled 79 participants (39 m, 40 f) was 70.0 (64.0–76.5) years; the median duration of hospital stay was 17.0 (13.0–21.0) days. Median Charlson Comorbidity Index (CCI) and fragility index before admission to hospital in the studied group were seven and four, respectively. No detailed data were obtained from the hospitalization of 25 patients, therefore in Table 1 they are presented for the group of 54 patients. During hospitalization, interstitial pneumonia was diagnosed in 83.3% of them; 44% required supplemental oxygen therapy. Table 1 shows the detailed characteristics of the study population.

### 3.2. Self-Reported Symptoms (SRSQ)

At the time of evaluation M3, only 5/79 (6.3%) of patients were completely free of any COVID-19-related symptoms, while 57/79 (72.1%) reported the persistence of three or more symptoms. The most common symptoms were fatigue or muscle weakness 48/79 (60.76%), palpitations 32/79 (40.51%), nausea 32/79 (40.51%) and hair loss 24/79 (30.38%). 

Six months after discharge, at M6, 14/73 (19%) of patients reported a complete lack of symptoms, while still 39/73 (53.4%) had three or more persistent symptoms. The most common symptoms were fatigue or muscle weakness 35/73 (47.94%), palpitations 22/73 (30.14%), sleep difficulties 21/73 (28.77%), and nausea 20/73 (27.40%). Significantly fewer patients reported the presence of diarrhea and lower-grade fever than reported at M3. Most of the symptoms persisted or were even aggravated (chest pain, joint pain). Details are presented in Table 2.

Significant weight loss was observed (Friedman ANOVA: *p* < 0.001); the weight of 51 (64.6%) and 47 (59.5%) patients was lower at M3 and M6 respectively than before COVID-19 (Table 3).

### 3.3. Self-Reported Dyspnea (mMRC)

The presence of dyspnea symptoms before COVID-19 of a grade at least 1 (short of breath when hurrying on the level or up a slight hill) and of at least 3 (significant dyspnea) were reported retrospectively by 17/79 (21.52%) and 3/79 (3.38%), respectively. 

At the time of M3 evaluation—dyspnea symptoms of a grade at least 1 were significantly more frequent and were reported by 34/79 (43.03%) while 13/79 (16.46%) of patients had significant dyspnea with a score of at least 3. 

At the time of M6 evaluation—dyspnea symptoms significantly reduced compared to M3 but were significantly still more frequent than before COVID-19 (*p* = 0.015); 25/73 (34.25%) patients reported dyspnea symptoms with a grade of at least 1, and 6/73 (8.22%) had significant dyspnea with a score of at least 3. Details are presented in Table 3.

### 3.4. Health Related Quality of Life (EuroQoL)

Patients before COVID-19 were characterized by significant problems with mobility and usual activity. Only 38/79 (48.1%) and 40/79 (50.6%) reported no problems in these domains of EQ-5D-5L before hospitalization. As presented in Table 4, the decrease in quality of life at M3 affected all five domains of the EQ-5D-5L questionnaire, and most patients did not return to the pre-disease state at M6. The pain/discomfort dimensions were the ones most commonly impaired. At the time of the M6 evaluation, even more patients felt anxious or depressed than before COVID-19 (*p* = 0.042). 

The mean EQ-VAS score was 69.9 ± 17.6 before COVID-19, and significantly deteriorated to 61.1 ± 18.5 at M-3 (*p* < 0.001). At M6, the EQ-VAS score improved to 64.4 ± 16.2 compared to M3 but was still significantly lower than before COVID-19 (*p* < 0.001) (Table 4). 

### 3.5. Mortality

The course of COVID-19 was serious in the majority of patients; 83.3% suffered from pneumonia and 48% of them required oxygen therapy. Some patients received plasma of convalescents (16.7%), remdesivir (11.2%) and corticosteroids (44.4%). Only two patients were asymptomatic, and only seven did not suffer from pneumonia. In-hospital mortality in the studied group was 30.1% (62 out of 206 patients). Three-month mortality was 37.4% (77 out of 206 patients). In the next three months, an additional four patients died, bringing the overall mortality rate at M6 for the cohort to 39.3% (81 out of 206). The main direct causes of death during hospitalization were as follows: 23 sudden cardiac, 17 respiratory failure, 16 inflammation, 3 thromboembolic, 2 hemorrhagic, 1 heart failure. Within the first three months after discharge, 15 patients had died; 12 deaths were due to cardiovascular causes (nine sudden cardiac deaths, two exacerbations of heart failure, one mesenteric embolism); three deaths were due to various inflammation-related causes. Between the third and sixth month after discharge, four additional patients had died due to cardiovascular causes (three sudden cardiac death, one thromboembolic).

## 4. Discussion

To our knowledge, this is the first study to show the long-term consequences of COVID-19 in HD patients. Our data confirms that, unfortunately, serious sequelae of the disease extend beyond the hospital period, similar as in the general population. Three months after recovery, only 6.3% of HD patients were completely free of any COVID-19-related symptoms. Although significant improvements in this regard were found between three and six months of follow-up; many recovered HD patients still suffer from persistent symptoms. Indeed, the overwhelming majority of patients (81%) still experienced one or more symptoms six months after recovery, and 53.4% had three persistent symptoms or more. 

There is no reporting on the long-term consequences of COVID-19 in HD patients. Therefore, we need to relate our results to the increasing number of studies assessing the presence of post-COVID-19 symptoms in the general population. Prevalence of these conditions differs significantly between studies due to the different target populations, case definitions and diagnostic procedures used. Most of the early data emerged from the follow-up of hospitalized individuals who had a more severe disease course, and consequently reported a higher prevalence of persistent symptoms. For instance, in PHOSP-COVID, a multi-center UK observational study of adults discharged from hospital, 71% of participants did not feel fully recovered at a median of five months after hospital discharge [11]. Using the same standardized questionnaires as used in our study, Huang et al. showed in their prospective cohort study from Wuhan, China that 76% of convalescents reported at least one persistent symptom six months after symptom onset [7]. Quite recently, the same authors reported that one year after acute infection, COVID-19 survivors still had lower health status than did non-COVID-19 controls matched for age, sex, and comorbidities [12]. In a recent meta-analysis of 21 studies consisting of 47,910 patients (hospitalized and not hospitalized), 80% of patients with a confirmed COVID-19 diagnosis were reported to continue having at least one symptom beyond two weeks following acute infection [13]. In another meta-analysis the sample included 15,244 hospitalized and 9,011 non-hospitalized patients. The results showed that 63.2%, 71.9% and 45.9% of the sample exhibited ≥one post-COVID-19 symptom at 30, 60, or ≥90 days after onset/hospitalization [14]. On the other hand, the prevalence of symptoms that remain 12 weeks after SARS-CoV-2 infection in patients non-hospitalized may be as small as 3.0%, based on tracking specific symptoms (tighter definition), to 11.7% based on the wider questionnaire which includes the questions about the 21-symptom list that was recently reported from 26,922 UK residents by The Office for National Statistics [15]. Numerous studies report intermediate prevalence rates between the above-mentioned ones [16,17]. The results clearly indicate that age, being female, poor general health, pre-existing health conditions, high viral load, obesity or being overweight and white ethnicity were associated with higher risk of long COVID in the majority of the studies [7,11,12,15]. 

In our study, fatigue or muscle weakness and sleep difficulties were very common at six months after discharge. This is consistent with data from the general population both after SARS and SARS-CoV-2 [7,18]. Although our cohort reported feeling more breathless as compared to the period before infection, it was not the dominant persistent symptom as in other studies [19]. Unexpectedly, palpitations were reported very often in our cohort as the second most common persistent symptom; this deserves further attention. In the light of suspected cardiac arrhythmias, detailed cardiological diagnostics seem necessary, especially considering that a significant percentage of convalescents in our cohort died in the early period after discharge due to suspected sudden cardiac death. It is a well-recognized fact that COVID-19 may affect the cardiovascular system, and post-recovery can lead to myocarditis, arrhythmias, and thromboembolic events in subjects from the general population with and without preexisting cardiovascular disease [20]. Sudden cardiac arrest has also been reported in a few patients recovered from COVID-19 [18,21,22]. It cannot be ruled out that a similar situation occurs in HD patients who are very burdened with cardiovascular diseases. 

To the best of our knowledge, no studies have yet been conducted on the long-term complications of COVID-19 in patients with chronic kidney disease. So far, only one study has been published on the post-COVID outcomes in kidney transplant recipients (KTR). In the prospective cohort study by Basic-Jukic et al., only 11.53% of 104 KTR who survived acute mild to moderate COVID-19 had no clinical symptoms or were free from any laboratory abnormality during the median follow-up of 64 days (range: 50–76 days) after recovery. Persistent symptoms, with the most frequent being shortness of breath, were present in 45.2% KTR, while 71.2% individuals had one or more laboratory abnormalities. Six months after acute COVID-19, most of them significantly improved and had no symptoms. On the other hand, many patients from this study required rehospitalization for severe complications [23]. In our own findings, post-COVID-19 syndrome was observed in 70.1% of KTR and 26.9% of them reported at least three persistent symptoms. The most common symptoms were fatigue or muscle weakness (43.3%) followed by hair loss (31.3%), memory impairment (11.9%) and muscle aches and headaches (11.9%) (data unpublished yet).

Compared to the general population, HD patients have a generally low health-related quality of life, which in particular relates to a decrease in mobility or pain/discomfort dimensions [24,25]. This is also confirmed by the very frequent use of painkillers and supplements in this population [26,27]. The characteristics of the studied patients confirm this regularity. High CCI and a fragility index of 4 in our patients reflects their vulnerability. They are susceptible to damage in everyday life, with ailments limiting activity, and commonly report weakness and the need to rest during the day. COVID-19 leads to further deterioration in their clinical condition and quality of life, and this condition persists for three months after discharge. Lowering the quality of life affects all domains of EQ-5DL, but the highest increase in health problems was reported for the pain/discomfort domain. A worrying find is only a slight improvement in their quality of life after six months of follow-up; this quality did not come back to the pre-COVID-19 level, and the anxiety–depression dimension was also observed to further deteriorate. Attention should also be paid to weight loss in most of the patients studied. Chronic fluid overload has been identified as an independent predictor of mortality in HD patients, and 30% remain fluid overloaded at dry weight [28]. Achieving optimal fluid balance is therefore one of the central challenges in routine dialysis practice, while weight loss after COVID-19 may further exacerbate the degree of overhydration. Taking into account the obtained results, we recommend a body composition method (BCM) guided dry weight assessment in all HD patients after COVID-19 [29]. 

This study has several limitations. First, the study was an uncontrolled cohort study that precludes comparison of the frequency of outcomes with patients not suffering from COVID-19. Second, a significant percentage of patients refused to participate in the study, which likely introduced selection bias. Included participants may be different from those who were not included for a variety of reasons, such as being more motivated to participate because of unresolved symptoms. On the other hand, it cannot be ruled out that the severe and long course of the disease may have discouraged additional contact with medical staff outside of dialysis sessions. Fortunately, we recorded no significant difference in primary baseline characteristics (age and gender) between patients who were included in final analysis and those who were not. Third, because participants were asked to report health-related quality of life and dyspnea symptoms prior to COVID-19 in the third month following discharge, measurement bias cannot be ruled out. Fourth, missing hospitalization data and the low sample size precluded the investigation of factors associated with the presence of long-COVID-19. Fifth, this study may have obtained less accurate information compared to face-to-face communication or physical examination, mainly because of the nature of telephone follow-up. On the plus side, this is the first study to analyze the long-term sequelae of SARS-COV-2 infection in the dialysis population. We used questionnaires that are widely-used and well-validated [7]. Due to the fact that during the first wave of the pandemic in our region, almost all COVID-19 HD patients from the Pomeranian voivodeship were hospitalized and dialyzed in our hospital, the findings may be considered representative of those with mild to severe disease. Considering all of this, our findings should be interpreted as exploratory and need to be validated in future controlled studies with a larger sample involving HD patients with varying severity. Clinically, it is also meaningful to study the association between the post-COVID complications and the risk of early and late death after recovery. The design and sample size of our study did not allow such analyses.

In conclusion, our study showed that HD patients hospitalized due to COVID-19 have high in-hospital and after discharge mortality. This is related to their complex comorbidities and fragility. COVID-19 survivals experience long lasting COVID-19 related symptoms which can further reduce their already poor health-related quality of life. Such self-reported symptoms as weakness, palpitation and chest pain require further evaluation as they may reflect serious health problems. Our results highlight the need for a long-term follow-up of HD patients for diagnostic and rehabilitation programs [30,31].

## Figures and Tables

**Figure 1 jcm-10-04451-f001:**
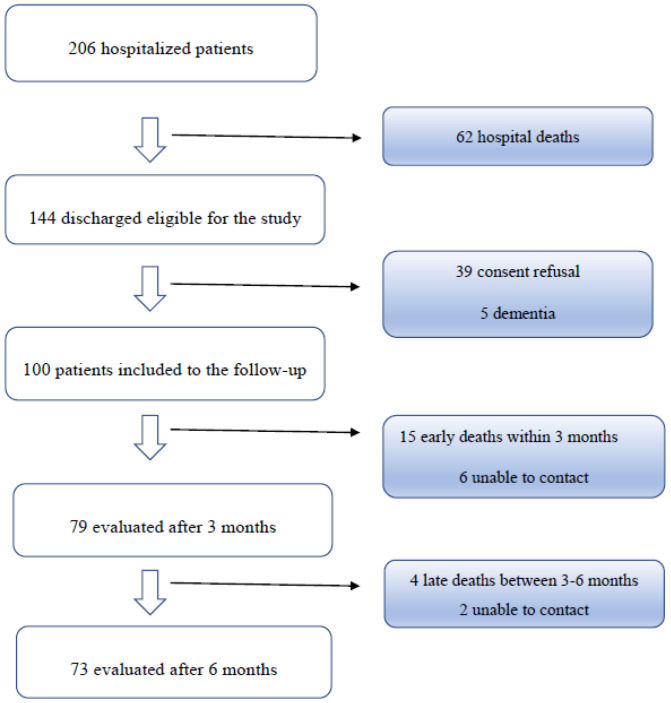
Flow chart of HD patients with COVID-19 admitted to the 7th Navy Hospital between 6 October 2020 and 28 February 2021.

**Table 1 jcm-10-04451-t001:** Characteristics of study patients.

*n*	79
Age years	70 (64–76.5)
Male gender	39 (49.37)
BMI kg/m^2^	26.1 (21.6–28.9)
Hemodialysis vintage months	40 (17.5–88)
Charlson Comorbidity Index	7 (6–9)
Fragility index	4 (3–5)
Diabetes mellitus	40 (50.63)
Arterial hypertension	74 (93.7)
Total duration of hospitalization days	17 (13–21)
Disease severity *:	
COVID-19 asymptomatic	2/54 (3.7%)
COVID-19 without pneumonia	7/54 (16.7%)
COVID-19 pneumonia	45/54 (83.3%)
COVID-19 with pneumonia requiring oxygen therapy	24/54 (48.1%)
Specific treatments *:	
Plasma of convalescents	9/54 (16.7%)
Remdesivir	6/54 (11.1%)
Corticosteroids	24/54 (44.4)

*—data only from 54 patients.

**Table 2 jcm-10-04451-t002:** Self-reported symptoms in SRSQ questionnaire.

Declared Persistent or New Symptoms	3 Months *n* = 79 *n* (%)	6 Months *n* = 73 *n* (%)
Fatigue or muscle weakness	48 (60.76)	35 (47.94) *
Palpitations	32 (40.51)	22 (30.14) *
Nausea	32 (40.51)	20 (27.40) *
Hair loss	24 (30.38)	18 (24.66) *
Decreased appetite	22 (27.85)	15 (20.55) *
Sleep difficulties	21 (26.58)	21 (28.77) *
Myalgia	20 (25.32)	15 (20.55) *
Dizziness	20 (25.32)	15 (20.55) *
Vomiting	20 (25.32)	14 (19.17) *
Headache	17 (21.52)	17 (23.29) *
Low grade fever	16 (20.25)	6 (8.22) **
Diarrhea	16 (20.25)	6 (8.21) **
Joint pain	14 (17.72)	16 (21.92) *
Taste disorder	13 (16.46)	8 (10.96) *
Smell disorder	12 (15.19)	7 (9.59) *
Chest pain	11 (13.92)	13 (17.81) *
Sore throat or difficult to swallow	10 (12.66)	6 (8.22) *
Skin rash	2 (2.53)	2 (2.74) *

*—non significant difference between 3 months vs. 6 months; **—*p* = 0.035 between 3 months vs. 6 months.

**Table 3 jcm-10-04451-t003:** Self-reported dyspnea (mMRC questionnaire) and patient weight before COVID-19, 3 months and 6 months after discharge.

	Before	3 Months	6 Months	*p* Value
mMRC score	1.42 ± 0.84	1.99 ± 1.25	1.67 ± 1.05	*p* < 0.001
mMRC score ≥ 1	17/79 (21.52%)	34/79 (43.03%)	25/73 (34.25%)	*p* = 0.015
Patient weight kg	75.7 ± 18.4	72.5 ± 16.6	72.8 ± 17.3	*p* < 0.001

**Table 4 jcm-10-04451-t004:** Patients with no problems with quality of life in five dimensions of EQ-5D-5L and health related quality of life in EQ-VAS questionnaire.

	Before COVID-19	3 Months	6 Months
Mobility	38 (48.1)	29 (36.7)	28 (38.4)
Self-care	58 (73.5)	52 (65.8)	50 (68.5)
Usual activity	40 (50.6)	33 (41.8)	37 (50.7)
Pain or discomfort	62 (78.5)	47 (59.5)	52 (71.2) *
Anxiety	67 (84.8)	58 (73.4)	53 (72.6) ***
EQ-VAS 0–100	69.9 ± 17.6	61.1 ± 18.5	64.4 ± 16.2 **

* *p* = 0.032 (before vs. 3 months vs. 6 months); ** *p* < 0.001 (before vs. 3 months vs. 6 months); *** *p* = 0.042 (before vs. 6 months).

## Data Availability

Detailed data are available on request from corresponding author.

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
