# Peer review of "Persistent Post-COVID-19 Syndrome in Hemodialyzed Patients—A Longitudinal Cohort Study from the North of Poland"

_jcm, 2021, doi:10.3390/jcm10194451_

Round 1

Reviewer 1 Report

The paper concerns of a very important problem - so called long-covid ( or post-covid syndrome) and has potential to be cited.

Although the examined group is rather small there is no similar paper on hemodialysis patients published so far, and only one similar paper concerning kidney transplant recipients.

The limitations of the study are properly addressed .  

Data a presented properly, however the discussion could be improved – only 17 paper are cited (among them 3 are the authors’ own papers) . More than 20 papers on long-covid were published within last two months.  The discussion part should be re-written to compare authors’ data to the long-covid symptoms of both  general population and chronically ill patients.

Author Response

Dear Editors,

Thank you for reviewing our manuscript. According to the Reviewers recommendations we made some changes in the manuscript. Below we present a summary of the changes made and responses to the comments of Reviewers. Changes made are marked in red in the manuscript.

REVIEWER 1

Only 17 paper are cited. More than 20 papers on long-covid  were published within last two months. The discussion part should be re-written to compare authors data to the long-covid symptoms of both general population and chronically ill patients.

Following Reviewer suggestions the discussion was expanded with the latest reports about the post-COVID syndrome. In particular, the topic of long-COVID prevalence and differences in outcome was discussed. To date, no study of persistent symptoms in patients with chronic kidney disease has been published. The only study conducted in the population of kidney transplant recipients was discussed in details, as well as our own experience on this point. The list of references currently includes 31 items. Changes made in the present version of the manuscript are marked in red

Reviewer 2 Report

The aim of this cohort study is based on post-COVID symptoms collected by phone in the 3rd and the 6th month after COVID-19 infection in the north Poland (Gdansk Hospital). Some parts of the manuscript are very well-written; however, a deeper literature review is required. Clinically, it is meaningful to study the association between the post-COVID complications in dialyzed population together with the decreasing risk in morbidity as well as mortality in this fragile population; conversely, a definitive sample was really small: only 73 cases were interviewed in the follow-up. However, only 54 of them were included into the descriptive statistics about the severity of COVID and its therapy (without specifically explaining why the number changed).

  1. the title described that persistent post-COVID-19 syndrome indicates an increased risk of serious health problems, yet the analyses of the paper only showed changes between baseline and follow-up in variables by bivariate analyses. Claiming changes in risk factors need to be shown by the regression model. Updating statistical analyses (such as mentioned in the previous sentence) is needed.

It is expected that the baseline and follow-up change in time for one variable (patient perception or the symptoms). Important is whether there has been or not a statistically significant associations to increase the post-COVID-19 risk.

2.The authors certainly know that BMI is not sufficient information for dialyzed patients. BCM is more adequate (this method is a more valid diagnostic measurement to estimate the correct optimal weight in HD/PD responders).

  1. The analyses between respondents and non-respondents at baseline and follow-up regarding age, gender, and comorbidities or CCI might be useful to show no selection bias or to be discussed in the discussion part.
  2. Please add into Table 1 two additional columns: data from medical records regarding to “time zero” and be more specific in M3 and M6 (baseline and follow-up).

Author Response

Dear Editors,

Thank you for reviewing our manuscript. According to the Reviewers recommendations we made some changes in the manuscript. Below we present a summary of the changes made and responses to the comments of Reviewers. Changes made are marked in red in the manuscript.

REVIEWER 2

  1. Deeper literature review is required. Clinically, it is meaningful to study the association between the post-COVID complications in HD patients with risk of mortality.

The discussion was expanded with the latest reports about the post-COVID syndrome. In particular, the topic of long COVID prevalence and differences in outcome was discussed. The necessity to undertake research on the associations between long-COVID persistent symptoms  and late mortality in COVID survival was emphasized as follows:

„Considering all of this, our findings should be interpreted as exploratory and need to be validated in future controlled studies with a lager sample involving HD patients with varying severity. Clinically, it is also meaningful to study the association between the post-COVID complications and the risk of early and late death after recovery. The design and sample size of our study did not allow such analyzes.

  1. Only 54 of them were included into the descriptive statistics about the severity of COVID and its therapy without specifically explaining why the number changed.

It was clarified in the results section as follows:

no detailed data were obtained from the hospitalization of 25 patients, therefore in Table 1 they are presented for the group of 54 patients”.

  1. The title described that persistent post-COVID-19 syndrome indicates an increased risk of serious health problems, yet the analyses of the paper only showed changes between baseline and follow-up in variables by bivariate analyses. Claiming changes in risk factors need to be shown by the regression model. Updating statistical analyses (such as mentioned in the previous sentence) is needed. It is expected that the baseline and follow-up change in time for one variable (patient perception or the symptoms). Important is whether there has been or not a statistically significant associations to increase the post-COVID-19 risk.

We agree with the Reviewer that the title of the manuscript incorrectly suggests that its purpose was to analyze predictors of late serious complications, e.g. mortality. The purpose of the study, however, was to assess the frequency of specific persistent symptoms of COVID-19 after 3,6 months. Therefore, in the current version of the manuscript, the title of the study has been shortened / changed as follows: “Persistent post-COVID-19 syndrome in hemodialyzed patients - a longitudinal cohort study from the North of Poland.

As stated in the method section our outcome measures were: 1) the percentage of patients with persistent of COVID-19 symptoms in SRSQ;  2) mMRC score  ≥1 in mMRC scale; 3) the percentage of responders reporting no (not any) problem across each of the five EQ-5D-5L dimensions; 4) quality of life in the analog EQ-VAS scale. The sample size was too low to perform reliable analysis of predictors of late poor outcomes e.g. mortality.  Given 15 cases (late deaths), it was difficult to assess even two predictors in the multivariable regression analysis. Besides, we did not have the exact causes of late deaths to associate mortality with complications of COViD-19. However, the necessity to undertake such studies was emphasized in discussion.

  1. The authors certainly know that BMI is not sufficient information for dialyzed patients. BCM is more adequate (this method is a more valid diagnostic measurement to estimate the correct optimal weight in HD/PD responders).

We agree with the Reviewer's opinion and a relevant comment on this point has been added in the discussion as follows:

“Chronic fluid overload has been identified as an independent predictor of mortality in HD patients, and 30% remain fluid overloaded at dry weight [28]. Achieving optimal fluid balance is therefore one of the central challenges in routine dialysis practice while weight loss after COVID-19 may further exacerbate the degree of overhydration. Taking into account the obtained results, we recommend  to perform a body composition method (BCM) guided dry weight assessment in all HD patients after COVID-19 [29]”.

  1. The analyses between respondents and non-respondents at baseline and follow-up regarding age, gender, and comorbidities or CCI might be useful to show no selection bias or to be discussed in the discussion part.

 It was done as requested. We made a comparison of the data we have. The relevant information was added to the results and discussions sections as follows:

Results: “The excluded patients did not differ from the study group in terms of age: 68.5 (57-80.75) years and gender (51.43% males)”

 Discussion:Included participants may be different from those who were not included for a variety of reasons, such as being more motivated to participate because of unresolved symptoms. On the other hand, it cannot be ruled out that the severe and long course of the disease may have discouraged additional contact with medical staff outside of dialysis sessions. Fortunately, we recorded no significant difference in primary baseline characteristics (age and gender) between who were included in final analysis and those who were not”.

  1. Please add into Table 1 two additional columns: data from medical records regarding to “time zero” and be more specific in M3 and M6 (baseline and follow-up).

 Almost all data presented in Table 1 did not change during follow-up (eg, age, gender, comorbidity, history of dialysis, history of COVID-19 treatment). Moreover, they were not monitored. Therefore, table 1 has not been changed

We would like to thank the Reviewers for all comments. They allowed to raise the quality of our work.

Round 2

Reviewer 1 Report

All the concerns were properly anwered and corrected.